# Can't simply roll it out: Evaluating a real-world virtual reality intervention to reduce driving under the influence

**Daniel Vankov***, **Ronald Schroeter**, **Divera Twisk**

Queensland University of Technology (QUT), Centre for Accident Research and Road Safety–Queensland (CARRS-Q), Brisbane, Queensland, Australia

ᴑ These authors contributed equally to this work.

* d.vankov@qut.edu.au

**Data Availability Statement:** All relevant data are within the manuscript and its Supporting Information files.

**Funding:** The Australian Government Department of Education supported this work through a 2015

## Abstract

Driving under the influence (DUI) increases the risk of crashes. Emerging technologies, such as virtual reality (VR), represent potentially powerful and attractive tools for the prevention of risky behaviours, such as DUI. Therefore, they are embraced in prevention efforts with VR interventions primed to grow in popularity in near future. However, little is known about the actual effectiveness of such DUI-targeting VR interventions. To help fill the knowledge gap, this study explored the effects of one VR intervention as delivered in the real world. Using pre and post test design, including an intervention group (n = 98) and a control group (n = 39), the intervention evaluation examined young drivers' (aged 18 to 25, no known history of DUI) intention and self-reported behaviour three months after the intervention as compared to the baseline. The results did not provide evidence for statistically significant effects of the VR intervention on self-reported DUI behaviour during the three months post intervention and DUI intention at three months post intervention. Such results might be due to the fact that the recruited participants generally self-reported little DUI behaviour, i.e. positively changing behaviour that is already positive is inherently challenging. Nevertheless, the results question the utility of funding the roll-out of arguably attractive technologies without a thorough understanding of their effectiveness in particular settings. To improve the potential for future positive outcomes of such interventions, we provide suggestions on how VR software might be further developed and, subsequently, leveraged in future research to improve the likelihood for behavioural change, e.g. by collecting, analysing and presenting objective driving performance data. Alternatively, future endeavours might focus on participants with known DUI history and examine the effects of the VR intervention for this particular higher-risk group.

## Introduction

### Young drivers, risks and information technology

Driving risks prevention efforts yield unsatisfactory results, leaving the safety of young drivers in focus [1]. Young people continue to be overrepresented in road fatalities [2]. Driving under

Endeavour Postgraduate Scholarship for DV. The Australian Government Department of Education did not play any role in the study design, data collection and analysis, decision to publish, or preparation of the manuscript.

**Competing interests:** The authors have declared that no competing interests exist.

the influence (DUI) of drugs or alcohol impairs driving performance [3] and, as a result, increases the risk of crashes [4].

Drink-driving is identified as a primary contributor to 30% of the fatal and 9% of the non-fatal injuries, while drug-driving is a main behavioural factor in 7% of the fatal and 2% of the non-fatal crashes for all drivers in Australia [5]. Young drivers (18–29 years) report higher than expected engagement in DUI [6,7]. For instance, in the United States, 7.8% of the young driver reported driving after drinking alcohol during the month before they were surveyed, while 20% rode with a driver who had been drinking [8]. The risk of DUI increases when the driver is young [9], leading to a five-fold increase of subsequent DUI-related risks in comparison with older drivers [9].

To reduce serious crashes involving young drivers, DUI interventions [10,11] may leverage opportunities provided by technology [12]. A potentially higher added value to reduce road trauma amongst young drivers may be hidden in utilising emerging technologies, such as virtual reality (VR). The use of VR driving simulators for promoting young drivers' safer driving is commonly referred to in the literature [13,14]. For example, a search in Scopus with the term *(ALL (road OR driver) AND ALL (safety) AND ALL ("virtual reality"))* returns over 5,000 results. However, the contemporary understanding of VR assumes delivery through consumer-grade devices such as HTC Vive, Oculus Rift, PlayStation VR, Google Daydream View and Samsung Gear. Although young drivers use modern VR in their daily lives [15], studies that incorporate this contemporary understanding are rare and even more so in the domain of DUI [16].

Despite this shortage of evidence around contemporary VR interventions in the literature, such interventions to target DUI are growing in popularity and implemented across continents (S2 Appendix). For example, the "Virtual reality and real volunteers for Advancing Safer Traffic participation for young people" (VAST) project reports making available its VR technology to 5,301 young people in Argentina, Belgium, Bulgaria, the Netherlands and Romania (S2 Appendix). Reviews of the implemented VR interventions appeared in media (Internet, newspapers, radio, and TV) reportedly reaching another 200,000 people (S2 Appendix). To achieve all that, VAST received support from the European Union Erasmus+ program. Such support can amount to up to 150,000 EUR. Despite the prestigious support, the wide outreach and the positive reception of the VR intervention, also similar to other real-world road safety interventions [17], the project promoters did not succeed in assessing whether their intervention triggered a behavioural change in the participants (S2 Appendix).

By not evaluating behavioural changes, an opportunity was missed in VAST, and potentially in other similar initiatives, to generate a better understanding of the involved behaviour change processes, as well as to find out what kind of participants get involved in such interventions, e.g. whether they are likely to report engaging in DUI in the first place. To help fill that gap, following the CONSORT 2010 Statement [18], the current paper evaluated whether young drivers' self-reported past DUI behaviour and DUI intention alter after the exposure to a contemporary VR intervention, identical to the one implemented in VAST, in a three-month quasi-experiment.

## Related work

The VR market is estimated to grow from 10.32 billion USD in 2019 to 62.1 billion USD in 2027 or at a compound annual growth rate (CAGR) of 21.6% [19]. Looking at the education sector, exclusively, the forecasted CAGR until 2024 is estimated even higher, at 59% [20]. Not surprisingly, VR is becoming a popular tool with both practical and research applications [21].

There is some evidence of the VR's potential to treat clinical impairments [22]. Nevertheless, evidence of VR effectiveness when used in clinically-healthy populations is scarce. The

literature also provides support for using VR to increase empathy [23–25]. For example, as a bullying prevention tool among school students (n = 118), in the study intervention group, VR significantly increased the feelings of belonging and willingness to intervene [24]. Other interventions led VR-users to reduce their paper use [26], not to cross streets when texting [27] or to improve their diabetes self-management [28]. Thus, VR is seen as capable of triggering real-life behavioural change [29]. Nevertheless, such conclusions are based on studies with small intervention (e.g. max 29 participants) and control groups (e.g. max 29 participants, if present at all) [29].

VR offers a somewhat unique minimisation of treatment risks in safe training environments [22]. With associated risks being removed, in real-time, VR is capable of delivering performance feedback and cues to change behaviour as well as retrospective behaviour reviews and analyses [22]. Such opportunities can be valuable when targeting behaviours which can lead to life-threatening situations, such as DUI, the behaviour explored in this paper.

DUI is challenging to investigate and research under real-life conditions is limited due to both ethical and legal considerations [30,31], i.e. the behaviour is illegal to perform. Thus, VR may represent a safe way to simulate DUI in road safety research, replacing questionable methods such as letting young people drive intoxicated, e.g. on a closed circuit [31].

In the road safety domain, contemporary VR studies are rare and vary considerably in their characteristics. For example, with respect to participants, VR studies may typically involve young drivers without articulating it explicitly, except in Agrawal, Knodler [32]. The reported numbers of involved participants range from 17 [33] to 1,900 [34]. In terms of their focus and what was measured, some collected data on risk and safety perception [35], simulator sickness [33] and VR simulation user-friendliness [36] through self-report. Others focused on the participants' driving, looking at performance measures such as hazards anticipation [32], target detection [37] and fuel consumption [34]. Only one study reports safety benefits as a result of using VR, i.e. in training participants to anticipate latent hazards [32]. However, the authors did not investigate whether these beneficial effects are sustained in the long-term. As such, long-term effects of such road safety VR interventions are generally unknown. This presents a gap in the literature, which this paper aimed to address.

## Study aim and hypothesis

In a quasi-experiment, the current study aimed to assess whether a real-world VR intervention influenced self-reported *Past DUI behaviour* and *DUI intention* in a volunteer sample of young drivers (aged 18 to 25) without preliminary information about any known records of DUI. The intervention was a stand-alone effort, identical to the source VAST project (S2 Appendix), which means it did not include any other interactions (apart from being incentivised, see recruitment below) with the participants, such as targeted recruitment, pre-screening, briefings or group discussions. We hypothesised that three months after the intervention and relative to the Control group, the participants in the Intervention group would report significantly lower *DUI intention* as well as significantly lower *Past DUI behaviour*.

## Materials and methods

### Recruitment of participants

The Queensland University of Technology (QUT) Human Research Ethics Committee approved the current study prior to commencement (Approval Number 1800000214). It was implemented as a quasi-experiment with convenience Intervention and Control groups. Aimed at improving retention rates, participation in a random draw of 10x$50 gift vouchers at Time 1 (T1, baseline, July 2018, before the intervention) and 10x$100 gift vouchers at Time 2

(T2, approximately three months after T1, November 2018) was offered as an incentive to participate. An additional $10 gift voucher was offered only to the Intervention group participants for driving the VR simulator.

The Intervention group participants were recruited face-to-face at venues with free public access where the VR driving simulator was installed. To be eligible to participate, a participant had to be aged 18 to 25 with a valid driver's license and no history of seizures or epilepsy due to potential risks associated with the use of the VR headset while seated in a static position [38].

A pilot test was carried out to test the equipment and the recruitment procedure. It took place in a public space at the Brisbane Queen Street Mall on the 28th of May 2018.

To aid the main recruitment of young drivers, only spots with high student traffic at the QUT campuses were considered for the VR driving simulator setup, such as the QUT library lobbies or the QUT Cube. The QUT Garden Point Campus HiQ reception space, level 3 of P block, was eventually chosen as the most appropriate location that was also available at the time. It is an open access area with a constant flow of people. The main recruitment took place between 16th and 27th of July 2018, from 10:00 a.m. till 4 p.m., Monday to Friday.

In parallel with collecting data from the Intervention group participants, from 19th to 31st of July 2018, a social media campaign was implemented on Facebook to recruit Control group participants. The Facebook campaign was set to target people who were aged 18 to 25, resided in Australia, spoke English and possessed a driving license.

During recruitment, all participants completed the same online survey at T1. Participants were expected to perform self-screening if they meet the inclusion criteria to participate before they consent and complete the survey. Implied consent was required for all participants. Such was considered obtained after a participant went through the study information sheet, generated their anonymous identifier, and started completing the survey. The anonymous identifiers were generated by the respective participants themselves, as per a predefined formula, and included: day of birth, first letter of first name, first letter of family name and last two digits of mobile number (example 24DL08).

At T2, an invitation to complete the second survey was sent to all participants. Although the participants were not surveyed whether they drove a car after T1, at the time of study design, three months were considered enough for them to have had the opportunity to do so and DUI, the risky behaviour of interest [39].

## Apparatus and context

A Unity-based VR software was operated on a driving simulator console, consisting of Oculus Rift VR headset, driving seat, driving wheel Logitech G29 and a computer. The computer, a consumer-grade gaming desktop, had enhanced processing-power capabilities to be able to run the VR scenarios in full quality. The scenarios, happening inside the virtual environment, were visualised for the public on a large TV. The public consisted of other young people attending the venue who were potential study participants. In many occasions, they were also friends of the participating drivers. Thus, the TV screen facilitated the attraction of new participants while triggering unintended casual discussions around the driving performance of the current ones.

Although when present, such unintended casual discussions might have been distracting to the participants completing the task, no effort to control for this type of distraction was applied. It is not uncommon for young people to drive in the presence of others in their vehicles and to be commented on. Thus, the situation was considered representative of possible driving conditions, which aligned well with the experiment.

Nevertheless, it was considered that viewing a scenario before participation could potentially influence subsequent participants. Despite the fact that such priming effects, for example realising the level of reduced vision while under the influence of alcohol and adjusting head movements to compensate, may also appear when young people are on the road, an effort was made to mitigate potentially confounding the data by advising the viewers to try a scenario they have not seen when their turn comes. Such advice would be reiterated while the respective participant was introduced in detail to how to operate the simulator with the VR software.

### VR intervention procedure

Immediately after completing the T1 survey, the Intervention group participants were subjected to the VR simulation, which took no more than 15 minutes per participant (Fig 1). A representative of the research team was present at all time to facilitate the intervention and answer any questions participants might have had about the study or the VR simulation. The simulation allowed the participants to "drive stoned" in a safe virtual environment, and to learn about the dangers of mixing substance abuse and cars. The VR software did not allow for any data to be collected about the participants' driving performance.

During the VR simulation, the participants could choose their "high" and step behind the wheel of a virtual car to experience simulated intoxicated experiences. The simulated experiences reveal to the users how their perception of reality and, therefore, driving performance is affected.

Each VR simulation was preceded with making sure the respective participant was as comfortable as possible. First, the participant was invited to take a seat. Second, the distance between the seat and the Logitech G29 simulator pedals was adjusted to the participant's preference. Third, the VR headset was adjusted to fit the participant (Fig 2).

After the VR driving simulator was adjusted for a participant's maximum comfort, the VR software was started. Once started, the software would position the participant as sitting into the driver's seat of a four-seat light vehicle with a clear view of natural night scenery. When turning their head around, the participant will be able to see a 360-degree view both in the car and of the outside scenery. In the vehicle, there is an empty front passenger seat and an empty back seat. The virtual car driving wheel will turn simultaneously together with the Logitech

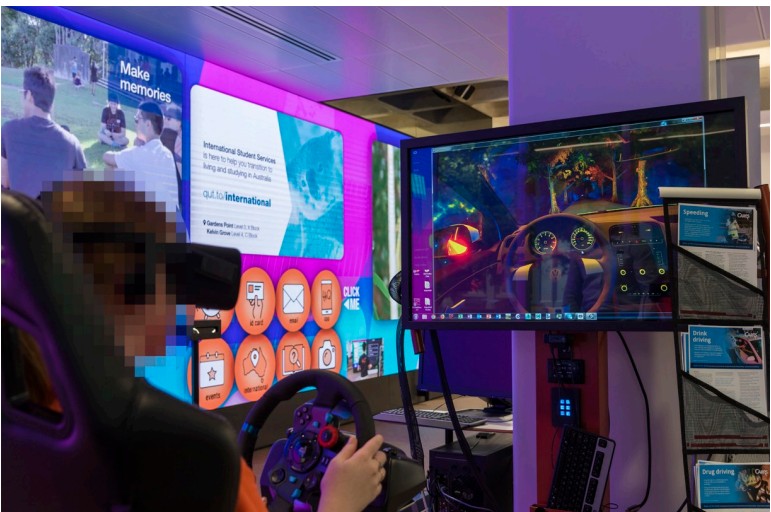

**Fig 1. A participant, operating the driving simulator with VR software.**

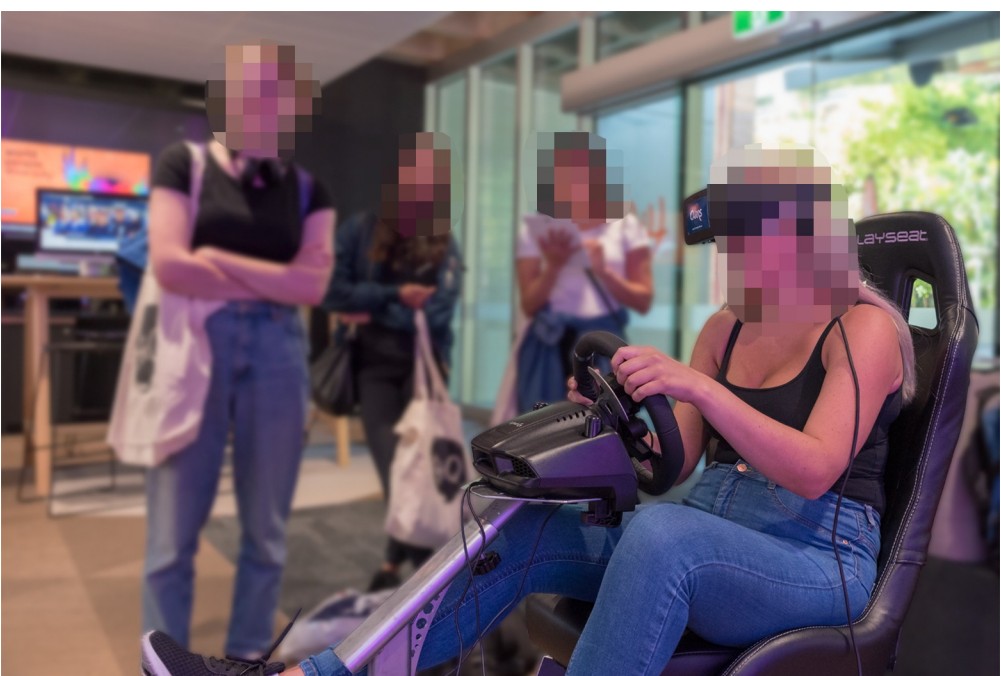

**Fig 2. The VR driving simulator, adjusted for a participant's maximum comfort.**

G29 driving wheel. A dashboard provides real-time feedback from the virtual car to the driver. Other feedback is received through the car mirrors, a big one in the centre of the windscreen and two at the sides. Generally, the interior looks like a real car. The only unrepresented thing is the participant, i.e. a participant cannot see their own body in the virtual environment. Upon the start of the software, the participant can hear the sound of turning the engine on. By default, the gear is put in automatic mode, ready to drive.

Initially, a participant would drive on a straight stretch of road without any other traffic participants. This experience was meant to give them time to get used to managing the VR driving simulator. It also served as the beginning of a plausible story, in which the participant was driving entirely sober to a night club (Fig 3).

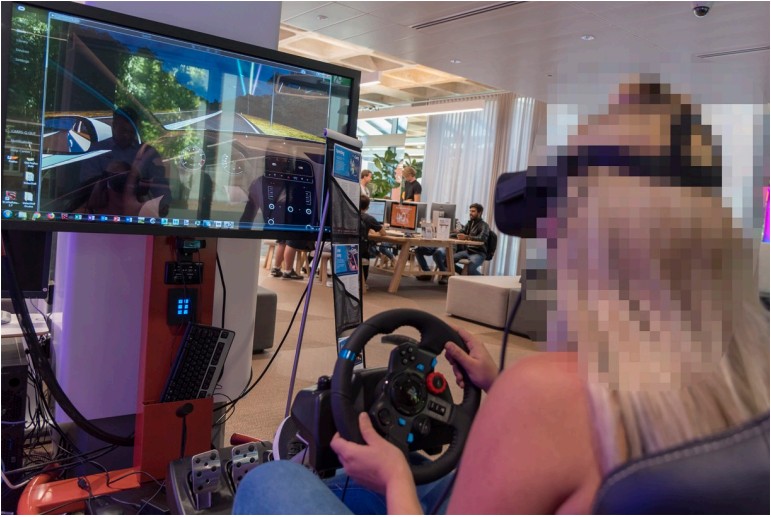

**Fig 3. A participant is getting used to managing the VR driving simulator.**

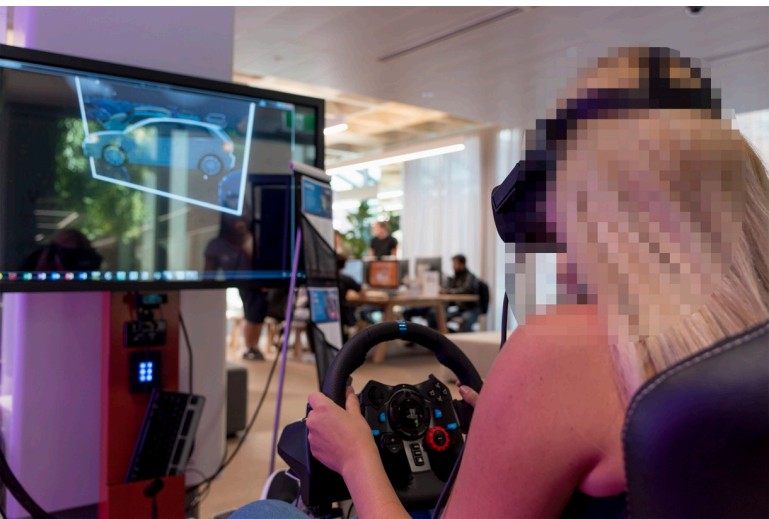

**Fig 4. The VR software visualises parking of the vehicle before entering the night club.**

At the night club, parking the vehicle and entering the premise was visualised (Fig 4).

Once inside the night club, each participant was given a choice between alcohol and drugs. The selection was made with the participant directing their look at their choice. If drugs were selected, then an additional option to choose between ecstasy, cannabis or magic mushrooms was given (Fig 5).

Once the final selection of what to be experienced was made, a picture with people on a dance floor was visualised. Subsequently, the participant would find themselves back into the car and ready to drive back home (Fig 6).

The VR was altering the subsequent driving experience, depending on the choice of substance. For example, in the case of:

- Alcohol, the vision area was reduced. There was a delay between the vehicle's response to a given command.

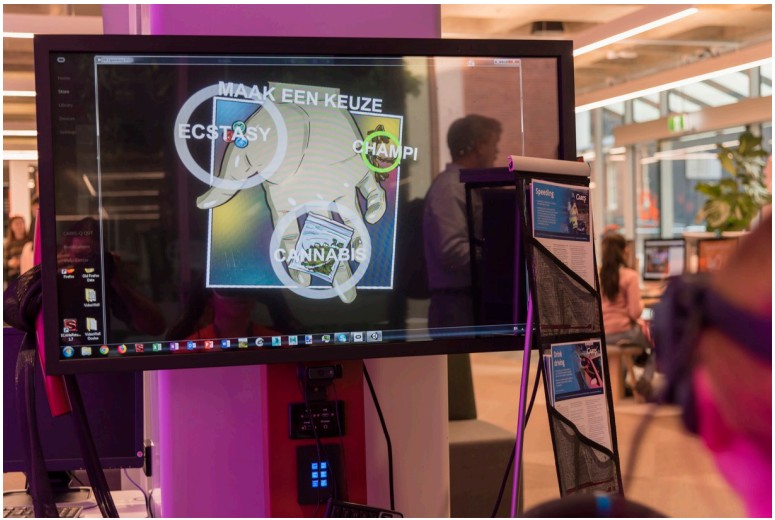

**Fig 5. A choice to experience impaired driving as a result of ecstasy, cannabis or magic mushrooms influence is given to a participant.**

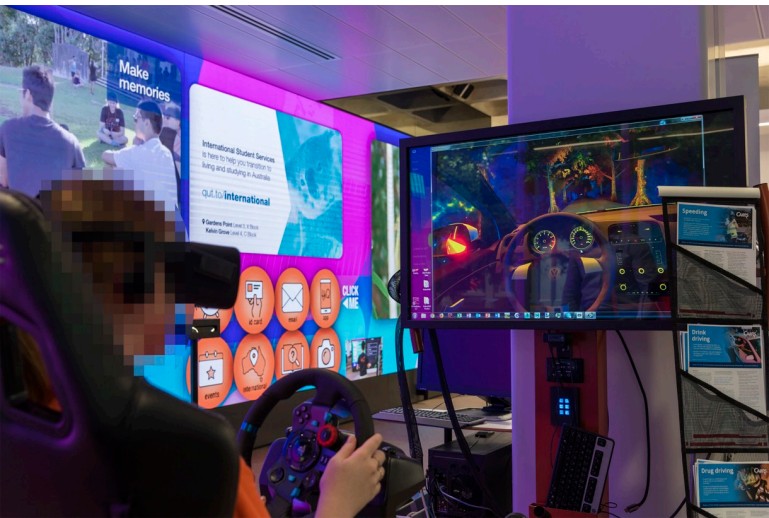

**Fig 6. A participant driving under the VR-simulated influence of magic mushrooms.**

- Ecstasy, everything moved at an increased pace. Sensors were sharpened. Everything was very colourful and flashy, also getting blurry at intervals.

- Cannabis, everything was very slow. Colours were calm. Vision did not stretch very far, very much the opposite of ecstasy.

- Magic mushrooms, the world was unreal, with imaginary sceneries and characters. The vehicle behaved opposite to the commands it received.

Different types of music additionally reinforced the chosen experience. Each experience was composed of several parts, happening on different types of roads, e.g. motorway or rural. If participants crashed, they were taken back to the beginning of the respective part, thus, being given a chance to correct their behaviour. The simulation was over when a point with a Police car was reached. Although there was no planned debriefing session after a participant completed their VR simulation, the research team members were answering all questions the respective participant might have had.

## Sample size

Initially, 282 participants took part in the VR intervention, and 70 completed the Control group survey (Fig 7). Partially completed surveys were not considered. After removing 23 entries with age "more than 25" and one duplicate case, 329 cases (72% male; Mage = 20.92 years, SD = 2.16; Mdrivingexperience = 3.25 years, SD = 2.07) were retained.

At T2, the 329 participants were contacted by e-mail to complete the second survey. 138 young drivers (66% male; Mage = 20.93 years, SD = 2.22; Mdrivingexperience = 3.38 years, *SD* = 2.07) completed the second survey (a 58% dropout rate), 39 Control group participants and 99 Intervention group participants. One Intervention group participant was subsequently excluded from the analysis as an outlier.

## The survey

The online survey consisted of five questions appearing in fixed order both at T1 and T2 (Table 1). Datasets from T1 and T2, related to the same person, were linked through the participants' self-generated anonymous identifiers.

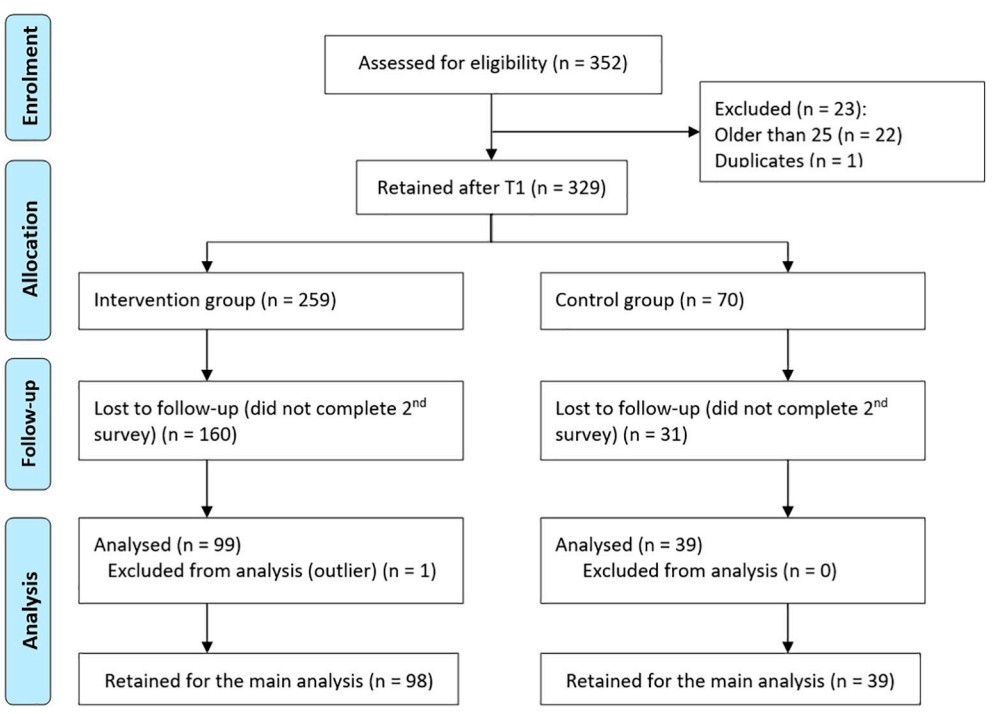

**Fig 7. Determining the sample size, CONSORT 2010 [18].**

## Analysis

The data from the surveys were checked, coded and entered into SPSS Statistics 25. Initially, descriptive statistics (means and standard deviations) were examined. To extract preliminary information about the relationships between the four main variables (*DUI intention* at T1, *DUI intention* at T2, *Past DUI behaviour* at T1, and *Past DUI behaviour* at T2), Spearman's rho correlation coefficients were calculated along with their significance levels.

Normality of *DUI intention* at T1 (skewness = 3.46, std. error = .21; kurtosis = 12.26, std. error = .41), *DUI intention* at T2 (skewness = 2.77, std. error = .21; kurtosis = 8.96, std. error = .41), *Past DUI behaviour* at T1 (skewness = 3.70, std. error = .21; kurtosis = 15.99, std. error = .41), and *Past DUI behaviour* at T2 (skewness = 3.91, std. error = .21; kurtosis = 17.67, std. error = .41), was assessed statistically. A visual examination of the histograms and the Q-Q plots as well as boxplots' outliers confirmed a departure from normality. The recommended choice in such situations is to use nonparametric tests, which rely on medians.

**Table 1. Measured constructs.**

| Category | Item | Question | Scale | Source |
|---|---|---|---|---|
| Demographics | Gender | How old are you (in years)? | *(whole number)* | N.A. |
| | Age | What is your gender? | Male / Female / Other | |
| | Driving experience | How much is your driving experience (in years)? | *(whole number)* | |
| DUI | DUI intention | How often do you think you will drive under the influence of alcohol or drugs in the next 3 months? | "Never" to "All the time" (1–9) | Adapted from Elliott and Thomson [40] |
| | Past DUI behaviour | How often did you drive under the influence of alcohol or drugs over the last 3 months? | "Never" to "All the time" (1–9) | |

**Table 2. Frequencies and percentiles per group condition (n = 137).**

| Variable | Dichotomised value | Intervention group (n = 98) | Control group (n = 39) |
|---|---|---|---|
| DUI Intention at T1 | Never | 88 (90%) | 36 (92%) |
| | Other than never | 10 (10%) | 3 (8%) |
| DUI Intention at T2 | Never | 79 (81%) | 30 (77%) |
| | Other than never | 19 (19%) | 9 (23%) |
| Past DUI behaviour at T1 | Never | 84 (86%) | 35 (90%) |
| | Other than never | 14 (14%) | 4 (10%) |
| Past DUI behaviour at T2 | Never | 86 (88%) | 30 (77%) |
| | Other than never | 12 (12%) | 9 (23%) |

The medians of the four variables were at the minimum of the scale, 1. Efforts to normalise the data through transformations (Lg10, Sqrt) did not satisfactorily improve the parameters. If non-parametric tests were applied to the data, the result would be that the intervention did not produce any effect. Thus, a decision was made to recode the two variables into categorical scales with two values, "0", denoting answers *other than never*, and "1", denoting a selection of the value 1 (*never*) as an answer, and to use non-parametric tests to analyse the data. The frequencies for the four variables (*DUI intention* at T1, *DUI intention* at T2, *Past DUI behaviour* at T1, and *Past DUI behaviour* at T2) are presented in Table 2.

A series of Chi-square tests for independence were run to investigate whether the Intervention group's frequencies of the answers *never* and *other than never* for each of those four variables were significantly different than the ones given by the Control group. Fisher's exact tests results were reported when the minimum expected cell frequency assumption for a Chi-square test for independence was violated. A series of McNemar's tests were performed to evaluate the effect of the intervention on the Intervention group's *DUI intention* and *Past DUI behaviour* over time to understand whether there was a significant change in the proportion of the answers *never* and *other than never* in the sample. McNemar's tests were also run on the Control group data to investigate whether their answers did not significantly change over time, either.

## Results

### Means, standard deviations and bivariate correlations

The Spearman's rho correlations between the variables were statistically significant in most of the cases (Table 3). For both groups, the correlations were strong between *DUI intention* and *Past DUI behaviour* both at T1 and T2, and moderate between *DUI intention* at T1 and *DUI intention* at T2. However, *Past DUI behaviour* at T1 was moderately significantly correlated with *Past DUI behaviour* at T2 only for the Intervention group (*rho* = 0.29, p < .001). Interestingly, the correlations did not reach statistical significance between *DUI intention* at T1 and *Past DUI behaviour* at T2 for the Intervention group (*rho* = 0.19, p = .056) as well as between *DUI intention* at T2 and *Past DUI behaviour* at T1 (*rho* = 0.26, p = .115) and between *Past DUI behaviour* at T1 and *Past DUI behaviour* at T2 (*rho* = 0.26, p = .108) for the Control group. Such results indicate that neither intention nor past behaviour might be a reliable determinant of future behaviour in the case of DUI.

### Between-group analysis

When investigating *DUI intention* at T1, the minimum expected cell frequency assumption for a Chi-square test for independence was violated. Fisher's Exact Test returned p = 0.75,

**Table 3. Frequencies, means, standard deviations and bivariate Spearman correlations (n = 137).**

| Group | Construct | Mean | SD | 1 | 2 | 3 | 4 |
|---|---|---|---|---|---|---|---|
| Intervention (n = 98) | 1. DUI intention at T1 | 1.12 | .39 | - | .30** | .64* | .19 |
|  | 2. DUI intention at T2 | 1.28 | .66 |  | - | .40** | .61** |
|  | 3. Past DUI behaviour at T1 | 1.16 | .45 |  |  | - | .29** |
|  | 4. Past DUI behaviour at T2 | 1.16 | .53 |  |  |  | - |
| Control (n = 39) | 1. DUI intention at T1 | 1.08 | .27 | - | .37* | .83** | .38* |
|  | 2. DUI intention at T2 | 1.33 | .70 |  | - | .26 | .76** |
|  | 3. Past DUI behaviour at T1 | 1.18 | .60 |  |  | - | .26 |
|  | 4. Past DUI behaviour at T2 | 1.39 | .88 |  |  |  | - |

* Correlation is significant at the 0.05 level (2-tailed).

** Correlation is significant at the 0.01 level (2-tailed).

indicating no significant association between the groups and their *DUI intention* at T1. A Chi-square test for independence (with Yates Continuity Correction) indicated no significant association between the groups and *DUI intention* at T2, either, $\chi^2$ (1, N = 137) = .06, p = .80, phi = .04.

A Chi-square test for independence (with Yates Continuity Correction) indicated no significant association between the groups and neither *Past DUI behaviour* at T1 ($\chi^2$ (1, N = 137) = .12, p = .72, phi = -.05) nor *Past DUI behaviour* at T2 ($\chi^2$ (1, N = 137) = 1.76, p = .19, phi = .14).

## Within-group analysis

The McNemar's test did show a statistically significant difference in *DUI intention* neither for the Intervention (N = 98, Exact Sig. = .06) nor for the Control (N = 39, Exact Sig. = .07) group before and three months after the intervention. The differences in *Past DUI behaviour* before and three months after the intervention for the Intervention (N = 98, Exact Sig. = .80) and the Control (N = 39, Exact Sig. = .18) group were not statistically significant, either.

Overall, those results did not show support for our hypothesis that three months after the VR intervention and relative to the Control group, the participants in the Intervention group would report significantly lower *DUI intention* as well as significantly lower *Past DUI behaviour*.

## Discussion

Through a quasi-experiment, the current study aimed to investigate if a VR intervention, implemented identically to wide-scope real-world interventions (S2 Appendix), produced any statistically significant changes over time in regards to the assessed *DUI intention* and *Past DUI behaviour*.

### Findings

The VR intervention was found to be able to influence neither the participants' *DUI intention* nor their *Past DUI behaviour* in the three-month study period. Separate tests did not find significant differences between the Control and the Intervention groups over time. The results were the same within the two groups.

These results are different than the observed safety benefits in a previous road safety VR study [32]. This difference might be due to the investigated behaviours, i.e. hazards

anticipation [32], while this study focused on DUI. Such results might be rooted in the inherent social unacceptability of DUI. In line with general expectations, participants mostly reported that they neither did DUI in the past nor intended to do so in the future (Table 2). Positively changing behaviour that is already positive is inherently challenging. Thus, before committing funding and rolling-out VR interventions, understanding who the potential participants are may increase the likelihood of triggering a behavioural change. In the future, it is suggested to look into the effect of the VR intervention for offenders. Alternatively, other constructs or different intervention settings [3,11] might be explored when implementing a VR intervention to influence DUI in a more general target group.

## Strengths

The current study was designed to evaluate a VR intervention's long-term effect on young drivers' DUI. The quasi-experiment took place in a safe environment, avoiding complicated arrangements if participants were to DUI an actual vehicle [31].

The study was implemented similar to interventions undertaken by road safety advocacy groups in their regular activities [17, S2 Appendix]. Such interventions are easily replicable once the initial investment of purchasing the necessary hardware and software is made. However, many such interventions are not robustly evaluated [17, S2 Appendix]. Thus, the collected self-reported data provided information that was not readily available from other sources.

Initially, an effort was made to involve a comparatively large sample as an Intervention group (n = 282) to improve generalisability of the findings in comparison with other lower-cost studies, focused on young drivers [41,42]. Despite the large drop-out rate, the retained sample (n = 137, control = 39) is considerably larger than the samples reported in previous studies with robust VR intervention evaluations [29]. A particular strength of this study was that a Control group was established to account for any general influence, which might have been experienced by the Intervention group participants. Such control is often missing in VR interventions studies [29].

## Limitations

Data was collected through online questionnaires, and self-reports are known to be inherently susceptible to bias. It has to be acknowledged that the investigated behaviour of DUI is considered socially unacceptable. Thus, higher pressure to report socially acceptable answers may be expected. The anonymous nature of the data collection, the impossibility of consequences for reporting DUI, as well as the fact that rewards were offered, irrespective of provided answers, should have minimised bias.

Data were collected twice, before and three months after the intervention. Thus, the evaluation of the long-term intervention effect did not incorporate the assessment of changes in the constructs of interest immediately after the intervention. It is acknowledged that with the participants being still present at the intervention venue, additional data could have been collected. However, at the time of evaluation design, this further data collection was considered as imposing extra time-consuming effort on the participants, an effort misaligned with the overall research focus on the longer-term effects. In such a situation, and to limit the potential drop-out rates, a decision was taken not to overburden the Intervention group participants. Instead, the study focused on the possible long-term effects only, which are much less often explored in the literature.

Another limitation of the study was the initial sample gender distribution (91 females, 236 males). Such distribution might not be a fair representation of the Australian young drivers'

population. An additional limitation of the sample was that the Intervention group was recruited predominantly on campus. Only a negligible number of 5 eligible participants was recruited during the pilot intervention. University campuses are a common source for study participants' recruitment [41,43], but it limits the generalisability of findings. For example, students in psychology often represent a significant proportion of the subjects in psychological research, which introduces a known bias in findings [44]. Although the study involved QUT students primarily, due to the choice of an intervention venue with free access, it can be assumed that the participants had diverse backgrounds and were pursuing different degrees in the university.

The study was limited by the number of participants who completed both surveys. Although the collected data from 137 participants at T2 was a comparatively larger sample than in previous VR studies [29], it was still insufficient for robust conclusions. Potentially, case-targeted measures to reduce drop-out rate may prove useful in future research, although this may reduce the real-world resemblance of the study.

Violations of assumptions were an observed data-related issue. The violations forced the application of non-parametric tests, limiting the validity of the findings.

A final limitation was that the perceived ease or difficulty of driving under the different conditions was not assessed for the individual participants. It is acknowledged that a participant might have found driving in the impaired VR simulation easy, or potentially more challenging and fun. In such a case, the intervention could have promoted DUI, despite the good intention behind its implementation. However, the intervention was delivered "as-is" in the real world and was evaluated as such. Furthermore, the data did not provide evidence for increased DUI three months after the intervention.

## Future research

To better understand the potential of VR to trigger behavioural change, future studies may look into leveraging the VR software to collect driving performance data. Similar to other VR applications [22], driving data could potentially be collected while the participants are experiencing DUI driving scenarios. Comparing experience both in "sober" and in DUI mode can be highly beneficial [22]. To enable such comparisons, the VR software developers can embed the same situations in both modes. For example, a situation that would trigger a participant's braking reaction can collect data about the time needed to react in DUI and compare it to the reaction time in "sober" mode. Such driving tasks can be more complicated, e.g. lane-keeping or overtaking, but can provide the participant with a realistic understanding of their abilities and how they are influenced when intoxicated [22]. Data can also be collected about obeying traffic rules or to the number of crashes and potential victims when DUI. Such a variety of quantitative data can serve as a basis of a much more in-depth evaluation of the drivers' performance under influence. By presenting the data to the respective driver, the overall experience may become more meaningful and informative [22]. Increasing the informative value of the experience can potentially influence the determinants of the participants DUI, such as their attitudes and norms [10]. Such a comprehensive influence on attitudes and norms may have a higher potential to trigger the desired behaviour change [10].

In summary, the potential of the VR as a tool should not be underestimated as it attracts both participants and public attention (S2 Appendix). However, supporting previous findings in the domain of DUI prevention for young drivers [10], to achieve positive behavioural change, an intervention may need to offer a broader VR experience than the replicated VAST experience (S2 Appendix). Software with enhanced capabilities to collect data and deliver feedback can help understand the full potential of using VR as a road safety tool in future research.

## Conclusion

Raising awareness of driving-related risks is not new, and education programmes have been studied in the past [45]. In such cases, researchers were able to provide essential insights into understanding the full intervention implications, and subsequently, suggest strategies for addressing gaps in their implementation. For example, researchers suggest that information technologies are potentially persuasive tools that might help young drivers adopt safer driving behaviour [46]. VR is one such technology. However, VR is a new tool. With the VR technology becoming increasingly available [15] and finding its way into prevention efforts (S2 Appendix), the question of how VR can potentially improve road safety is due for an in-depth investigation. At the same time, there is a limited number of VR studies in road safety. The literature review did not identify another VR quasi-experiment on DUI. Therefore, investigating the effects of the current VR intervention makes a unique contribution to the literature.

The current study assessed the impact of a quasi-experiment with VR software as an intervention tool. By doing so, it exhibited several strengths besides its real-world nature, such as a control for any general influence, and a focus on long-term effects. Nevertheless, the study had its limitations, e.g. potential for self-report bias, large drop-out rate, and problematic data. Despite those limitations, the study not only expanded the knowledge around using VR in road safety but extended the knowledge into the context of DUI.

Overall, this study found that the implemented VR intervention did not trigger a statistically significant shift in the participants' self-reported *Past DUI behaviour* and *DUI intention*. Nevertheless, we do not question the utility of VR interventions for DUI in general, but rather the utility of this specific VR intervention for DUI as currently implemented by being installed in a public space. In such open-public opportunistic interventions, it is unlikely that there will be preliminary information about whether individuals have a recorded history of DUI. Similar to the current study, such self-reported information may become available after the data is analysed. Obtaining such information in advance may be the key to unlocking the potential of VR interventions for DUI for triggering positive behavioural change. If such information is available before an experiment takes place, researchers might find it beneficial to focus the intervention on young drivers who report DUI.

The null results within the current study might be due to a large extent to the fact that the recruited participants generally self-reported little DUI behaviour. This objective outcome may question the utility of funding the roll-out of arguably attractive technologies without a thorough understanding of their effectiveness in particular settings. To overcome such shortcomings, a number of ways to increase the potential of VR as an intervention tool were identified, such as real-time data collection and feedback delivery. If implemented, those suggestions should help both researchers in future studies and practitioners in future applied projects. Thus, they should be considered before VR interventions are rolled-out on a large scale.

## Supporting information

**S1 Appendix. Raw data.**
(XLSX)

**S2 Appendix. VAST evaliation report.**
(PDF)

## Acknowledgments

The authors are grateful to Responsible Young Drivers Nederland for making available their VR software for this study.

## Author Contributions

**Conceptualization:** Daniel Vankov, Ronald Schroeter.

**Data curation:** Daniel Vankov.

**Formal analysis:** Daniel Vankov.

**Funding acquisition:** Daniel Vankov.

**Investigation:** Daniel Vankov, Divera Twisk.

**Methodology:** Daniel Vankov, Ronald Schroeter, Divera Twisk.

**Project administration:** Daniel Vankov.

**Resources:** Daniel Vankov.

**Software:** Daniel Vankov.

**Supervision:** Ronald Schroeter, Divera Twisk.

**Validation:** Daniel Vankov.

**Visualization:** Daniel Vankov.

**Writing – original draft:** Daniel Vankov.

**Writing – review & editing:** Daniel Vankov, Ronald Schroeter, Divera Twisk.

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
