## [Decision Letter · Decision Letter 0]

4 Feb 2021

PONE-D-20-38963

Can't simply roll it out: Evaluating a real-world virtual reality intervention to reduce driving under the influence

PLOS ONE

Dear Dr. Vankov,

Thank you for submitting your manuscript to PLOS ONE. After careful consideration, we feel that it has merit but does not fully meet PLOS ONE’s publication criteria as it currently stands. Therefore, we invite you to submit a revised version of the manuscript that addresses the points raised during the review process.

Your manuscript has been assessed by two acknowledged experts on the topic covered in the study. Overall, their comments are considerably positive, but some key amendments (some of them minor, some of them more relevant) are required from the authors before proceeding to consider the acceptance of the paper. Please try to address these comments and suggestions as good as possible, in order to summit the revised paper to a new round of reviews.

We look forward to receiving your revised manuscript.

Kind regards,

Sergio A. Useche, Ph.D.

Academic Editor

PLOS ONE

Journal Requirements:

3. We note that Figures 1, 2, 3 includes an image of participants in the study. 

Reviewers' comments:

Reviewer's Responses to Questions

**Comments to the Author**

1. Is the manuscript technically sound, and do the data support the conclusions?

Reviewer #1: Yes

Reviewer #2: Partly

2. Has the statistical analysis been performed appropriately and rigorously? 

Reviewer #1: Yes

Reviewer #2: Yes

3. Have the authors made all data underlying the findings in their manuscript fully available?

Reviewer #1: Yes

Reviewer #2: Yes

4. Is the manuscript presented in an intelligible fashion and written in standard English?

Reviewer #1: Yes

Reviewer #2: Yes

5. Review Comments to the Author

Reviewer #1: Hello Authors,

Thank you for submitting to PLOS ONE. Regarding your submission, I have the following recommendations to strengthen the paper details and add scientific merit. Please revise the paper and clarify the following sections:

Apparatus and Context (Page 9): Would the friends of the participants be distracting to the participants completing the task? Was there any effort to control for this type of distraction? Further the potential for student participants could be at risk of viewing the experiment details which could potentially confound the data? (priming effect). Finally, were the participants given a training tutorial of how to use the HMD?

VR intervention procedure (Page 9): Was an experimenter or PI present if the participant had questions?

VR intervention procedure (Page 9): Please describe the VR software and what were its capabilities? How were the scenarios presented using what platform? (e.g., Unity, Unreal). Was a standard desk computer used? Or a computer with processing power capable of running the scenarios? Were the participants debriefed in some manner following the final phase of the quasi - experiment?

Sample Size (Page 11): List females age range, Mean, Standard Deviation, and driving experience. Female biographical information may be listed and not made for the reader to infer.

Means, Standard Deviation and Bivariate Correlations (Page 15): Consider listing Spearman’s correlations in the body of text (first sentence). It is unclear.

Data transformation (Page 13): Completed test for normality and data transformation to justify using non-parametric statistics. This process must be applauded because you completed the correct steps for non-normal data analyses. (No action required here).

Reviewer #2: SUMMARY

This paper attempts to measure the effectiveness of a public VR DUI simulator as an intervention to reduce intent to DUI through a quasi-experiment involving adults 18-25 aged 18-25 with no history of DUI. The authors show no significant effect of the intervention on DUI intent and self-reported DUI behavior.

STRENGTHS

This paper is very thorough in defining and showing data, data analysis methods, and procedures. It is very clear what the authors did to achieve these results. The authors also succeeded in recruiting a large sample size, relative to many VR studies concerning road safety.

KEY WEAKNESSES

Participants are individuals that have no recorded history of DUI and generally self-report no DUI behavior. It appears flawed to use individuals that would not DUI in the first place to assess the quality of any intervention for reducing DUI behavior. For example, in Elliott and Thomson's work that the authors cite as a basis for the survey questions, they specifically target drivers with demonstrated speeding history when assessing the validity of their behavior model. It is hard to say whether they could have achieved the same results if all of their participants simply reported little or no intention to speed.

It is therefore difficult to use the given quasi-experiment to question the utility of VR interventions for DUI, as it appears that any intervention would be ineffective on a population that was not intending to exhibit the behavior in question. The authors indeed acknowledge this in their "Findings" section. Yet, in their abstract and conclusion, the authors do not do the same, creating the impression that there was some flaw in the VR intervention itself or their survey methodology that lead to the results.

CONCLUSION

Due to the above mismatch between what appears to be the primary cause for the null results and the claims the authors make about VR interventions for DUI, I recommend revising the paper to be more clear about this fundamental fact of the experiment. As a reader, I felt confused for most of the paper after reading in the abstract that the participants had no history of DUI, but then feeling like this fact went unaddressed for the majority of the paper thereafter. This paper would appear to fully meet the intention of this journal if the conclusions matched what appears to be a clear picture painted by the data.

OTHER TARGETS FOR REVISION

The authors performed a great bit more data analysis than the hypothesis and other content before the "Analysis" section would lead readers to expect. It would greatly clarify the intent and content of the paper if the authors informed the readers that they also intended to explore other correlations that could be found in the data.

6. PLOS authors have the option to publish the peer review history of their article (what does this mean?). If published, this will include your full peer review and any attached files.

Reviewer #1: No

Reviewer #2: No

---

## [Author Response · Author response to Decision Letter 0]

11 Feb 2021

As instructed by the Decision Letter, a rebuttal letter that responds to each point raised by the academic editor and reviewers is uploaded as a separate file labeled 'Response to Reviewers'.

---

## [Decision Letter · Decision Letter 1]

5 Apr 2021

Can't simply roll it out: Evaluating a real-world virtual reality intervention to reduce driving under the influence

PONE-D-20-38963R1

Dear Dr. Vankov,

We’re pleased to inform you that your manuscript has been judged scientifically suitable for publication and will be formally accepted for publication once it meets all outstanding technical requirements.

Kind regards,

Sergio A. Useche, Ph.D.

Academic Editor

PLOS ONE

Additional Editor Comments (optional):

Reviewers' comments:

Reviewer's Responses to Questions

**Comments to the Author**

1. If the authors have adequately addressed your comments raised in a previous round of review and you feel that this manuscript is now acceptable for publication, you may indicate that here to bypass the “Comments to the Author” section, enter your conflict of interest statement in the “Confidential to Editor” section, and submit your "Accept" recommendation.

Reviewer #1: All comments have been addressed

Reviewer #2: All comments have been addressed

2. Is the manuscript technically sound, and do the data support the conclusions?

Reviewer #1: (No Response)

Reviewer #2: Yes

3. Has the statistical analysis been performed appropriately and rigorously? 

Reviewer #1: (No Response)

Reviewer #2: Yes

4. Have the authors made all data underlying the findings in their manuscript fully available?

Reviewer #1: (No Response)

Reviewer #2: Yes

5. Is the manuscript presented in an intelligible fashion and written in standard English?

Reviewer #1: (No Response)

Reviewer #2: Yes

6. Review Comments to the Author

Reviewer #1: (No Response)

Reviewer #2: (No Response)

7. PLOS authors have the option to publish the peer review history of their article (what does this mean?). If published, this will include your full peer review and any attached files.

Reviewer #1: No

Reviewer #2: No

---

## [Editor Report · Acceptance letter]

7 Apr 2021

PONE-D-20-38963R1 

Can't simply roll it out: Evaluating a real-world virtual reality intervention to reduce driving under the influence 

Dear Dr. Vankov:

I'm pleased to inform you that your manuscript has been deemed suitable for publication in PLOS ONE. Congratulations! Your manuscript is now with our production department. 

Kind regards, 

on behalf of

Dr. Sergio A. Useche 

Academic Editor

PLOS ONE